# Accounting for Non-Detects: Application to Satellite Ammonia Observations

Evan White [1,2], Mark W. Shephard [1,*], Karen E. Cady-Pereira [3], Shailesh K. Kharol [1,4], Sean Ford [1,5], Enrico Dammers [6], Evan Chow [1,2], Nikolai Thiessen [1,5], David Tobin [7], Greg Quinn [7], Jason O'Brien [1] and Jesse Bash [8]

1 Environment and Climate Change Canada, Toronto, ON M3H 5T4, Canada
2 Faculty of Science, University of Waterloo, Waterloo, ON N2L 3G1, Canada
3 Atmospheric and Environmental Research (AER), Inc., Lexington, MA 02421, USA
4 Atmo Analytics Inc., Brampton, ON L6S 6L2, Canada
5 Faculty of Science, University of British Columbia, Vancouver, BC V6T 1Z4, Canada
6 TNO, Climate Air and Sustainability, 3584 CB Utrecht, The Netherlands
7 Space Science and Engineering Center (SSEC), University of Wisconsin-Madison, Madison, WI 53715, USA
8 US Environmental Protection Agency, Research Triangle Park, Durham, NC 27709, USA
* Correspondence: mark.shephard@ec.gc.ca

**Abstract:** Presented is a methodology to explicitly identify and account for cloud-free satellite measurements below a sensor's measurement detection level. These low signals can often be found in satellite observations of minor atmospheric species with weak spectral signals (e.g., ammonia ($NH_3$)). Not accounting for these non-detects can high-bias averaged measurements in locations that exhibit conditions below the detection limit of the sensor. The approach taken here is to utilize the information content from the satellite signal to explicitly identify non-detects and then account for them with a consistent approach. The methodology is applied to the CrIS Fast Physical Retrieval (CFPR) ammonia product and results in a more realistic averaged dataset under conditions where there are a significant number of non-detects. These results show that in larger emission source regions (i.e., surface values > 7.5 ppbv) the non-detects occur less than 5% of the time and have a relatively small impact (decreases by less than 5%) on the gridded averaged values (e.g., annual ammonia source regions). However, in regions that have low ammonia concentration amounts (i.e., surface values < 1 ppbv) the fraction of non-detects can be greater than 70%, and accounting for these values can decrease annual gridded averaged values by over 50% and make the distributions closer to what is expected based on surface station observations.

**Keywords:** non-detects; CrIS Ammonia Cloud Detection Algorithm (CACDA); ammonia; Satellite Detection; CrIS

## 1. Introduction

Measurements from any instrument have a minimum observable limit. Any values that are below this limit cannot be detected, and we will refer to them as non-detects. The existence of these non-detects complicates statistical analysis. When averaged over regions or periods, not accounting for these non-detects leads to high biases, as the averaged value is generated excluding values below the detection limit, which will typically be lower than the measured values. A number of procedures for handling non-detects have been used, including simply ignoring them [1], substituting in zero values [2], and substituting in values derived from the minimum detection limit [3]. More advanced methods identify measurements below the detection limit and substitute in values derived from a distribution of well-calibrated measurements [4–6], and this is the approach taken in this study. Accounting for non-detects in the satellite remote sensing observations of trace gases (e.g., ammonia ($NH_3$), formic acid (HCOOH), peroxyacetyl nitrate (PAN)) is important,

as these can include a significant number of atmospheric states where background levels dip below the sensor's detection limit. Non-detects in trace gas measurements are caused by a lack of signal in the spectral region used to detect the species; in more precise terms, they occur when the absolute value of the signal-to-noise ratio (SNR) for the measured species falls below 1. In this study, we propose an approach for determining and handling non-detects in the Ammonia ($NH_3$) data obtained from an infrared sensor, the Cross-Track Infrared Sounder (CrIS), which is deployed on several polar orbiters (SNPP, NOAA-20, and NOAA-21).

$NH_3$ is produced principally by fertilizer application and animal waste [7], but in some periods and regions, biomass burning [8] and automobile exhaust [9] are also significant sources. $NH_3$ concentrations are rising due to the increase of large-scale, intensive agricultural activities [10], which are often accompanied by greater use of fertilizers and concentrated animal feedlots. $NH_3$ is a significant precursor of $PM_{2.5}$ particles and thus contributes to poor air quality in many regions. Urban areas downwind of biomass burning events and/or with high volumes of traffic (e.g., Delhi or Los Angeles) are especially impacted by this trend. As stricter controls on other $PM_{2.5}$ precursors ($SO_2$ and $NO_x$) are implemented, $NH_3$ may become the limiting factor on $PM_{2.5}$ production. Furthermore, when $NH_3$ is deposited onto the ground or into bodies of water, it acts as a fertilizer and disrupts local ecosystems. Typical examples of these disruptions are algal blooms, which occur in the Gulf of Mexico and the Chesapeake Bay.

$NH_3$ is highly reactive and thus has a short lifetime on the order of hours in the boundary layer [11]. This leads to high variability in space and time. Concentrations can vary by orders of magnitude with distance from a strong source, such as Confined Animal Feedlot Operations (CAFO).

The identification and inclusion of non-detects reduce the potential for high bias in regions where the ammonia sources either vary (e.g., seasonal events, episodic events such as forest fires, etc.) or in general tend to approach zero. A more extreme example of this would be large forest fire plumes over northern latitude regions where there are no other significant sources of ammonia. In this case, if non-detects were not taken into consideration then weekly or monthly means over the region would include only a few days of the smoke source but not the days where ammonia levels were below the detection limit, leading to a high bias in results. Whereas, when non-detects are taken into consideration the average ambient ammonia concentrations are better represented over this region during this period. In general, accounting for non-detects is important as inaccuracies in $NH_3$ retrievals can cause issues in applications such as integrated satellite-derived emission [7], deposition estimates [12], and chemical transport model evaluations [13].

In this paper, we apply our approach for determining and handling non-detects to the CrIS Fast Physical Retrieval (CFPR) $NH_3$ product. This is a post-processing step after the CFPR retrieval is carried out, whose objective is to handle pixel-level retrievals where the CrIS signal from $NH_3$ is below the detection limit. This can be a result of either (i) clear-sky non-detect conditions where the atmospheric signal (mainly a function of ammonia concentrations) is below the detection limit of the sensor, or (ii) thick cloudy conditions (cloud optical depths > 1) where the satellite sensor cannot detect the ammonia below the clouds [14]. $NH_3$ from anthropogenic emissions is generally concentrated below the clouds near the surface due to short boundary layer $NH_3$ lifetimes [11]. Ignoring these pixels entirely, or assuming a low $NH_3$ value if they are cloudy, leads to biases in regional means. Here we use the Visible Infrared Imaging Radiometer Suite (VIIRS) CIMG product [15] to identify cloudy and clear CrIS observations; non-detects under thick clouds are rejected and non-detects under clear skies are assigned representative values. The representative values were developed using in-situ surface observations. This study analyzes the effects of accounting for non-detects in the CrIS-SNPP (CrIS1) $NH_3$ pixel level (Level 2) data from May 2012 to May 2021. This analysis includes comparisons between surface and satellite distributions, and demonstrates the spatial representation of the effect of applying the pixel-level non-detects globally, separated by season.

Here we define a source region as an area that has active and detectable ammonia sources in the time frame under consideration (e.g., day, season); conversely, a non-source region has no active sources at the measurement time. Some locations, such as the American Midwest, are source regions in the warm seasons, but not in winter. A background region is an area that usually has no active sources, but may experience sporadic large emission events, such as forest fires.

## 2. Data Sources

### 2.1. CrIS Ammonia Observations

This general methodology can be used for many different satellite retrieval methodologies, but here we demonstrate our approach for handling non-detects with CrIS $NH_3$ retrievals obtained from the CrIS Fast Physical Retrieval algorithm (CFPR), which is described in detail by Shephard and Cady-Pereira [16], with updates in Shephard et al. [17]. Briefly, CFPR uses an optimal estimation approach to retrieve ammonia profiles from CrIS radiances in the $NH_3$ ν2 band centered around 967.5 cm$^{-1}$. This $NH_3$ band is in the "atmospheric window", where the atmosphere is nearly transparent and thus is less impacted by interference from other species. Only a narrow range of the CrIS spectrum is used (962.50–968.75 cm$^{-1}$, or 10 channels). The optimal estimation process includes satellite-retrieved water vapor and temperature profiles in its forward model, which are obtained from the NOAA NESDIS-unique CrIS-ATMS product system (NUCAPS) products [18]. Thus, the retrieval can remove the effects of temperature and water vapor on the measured ammonia signal [16], and then determine how much $NH_3$ is needed to account for the residual in the $NH_3$ band, which can be greater than 1 K in brightness temperature over a scene with high $NH_3$ amounts and good thermal contrast. CrIS $NH_3$ retrievals typically have approximately one degree of freedom for signal (DOFS). By allowing the shape of the profile to vary, the CFPR algorithm captures the change in vertical sensitivity due to different atmospheric conditions.

CrIS is a Fourier transform spectrometer (FTS) launched by the U.S. NOAA and NASA on the Suomi National Polar-orbiting Partnership (S-NPP) satellite on 28 October 2011, NOAA-20 satellite on 29 November 2017, and NOAA-21 on 10 November 2022. The CrIS footprint is 14 km at nadir and the swath width is 2200 km. In this study, we focus on nine years of CrIS SNPP satellite observations, which are in a sun-synchronous low Earth orbit with approximate overpass times of 01:30 and 13:30 local solar time. Note that in the nine-year CrIS SNPP $NH_3$ dataset from May 2012 to May 2021 has missing observations between March 2019 and August 2019 due to instrument issues.

CrIS has a spectral resolution of 0.625 cm$^{-1}$ and low spectral noise (0.04 K at 280 K) in the $NH_3$ spectral region [19]. This long-wave band of CrIS also has the advantage of being identical for both normal spectral resolution (NSR) and full spectral resolution (FSR) radiance files. The high CrIS SNR (~1600), combined with the high thermal contrast associated with the early afternoon overpasses, provides enhanced sensitivity in the boundary layer where $NH_3$ is mostly concentrated. Even with this sensor sensitivity, there are significant numbers of observations where the atmospheric $NH_3$ signal falls below the detection limit of the sensor. We define the $NH_3$ SNR as the ratio of the online/off-line brightness temperature difference divided by the instrument noise in the $NH_3$ spectral region; if this ratio is less than 1, the NH3 signal is below the detection limit. The CrIS $NH_3$ detection limit in the infrared depends mainly on the amount of $NH_3$ and the surface temperature and its vertical profile, and ranges from ~0.3 to 1.0 ppbv. Currently, the uncertainty in the spatial and temporal variability of ammonia emission inventories is such that no reasonable prior assumptions are available from traditional sources (e.g., climatologies or chemical transport model fields that vary seasonally and by latitude and longitude). This is particularly true for background amounts as natural and small anthropogenic sources of ammonia are not well known. Therefore, the CFPR algorithm selects an a priori profile from three possible choices based on the estimated strength of the $NH_3$ signal [14,16]. Retrievals of minor species with a weak atmospheric signal (e.g., $NH_3$) in background conditions that

utilize radiative transfer forward model perturbations may fall into null-space (since the Jacobians approach 0), which often results in the retrieval not converging. Thus, in the CFPR approach if the signal falls below the noise then the observations are identified as a non-detect and the full retrieval, which can be computationally time-consuming and may provide no additional observational information, is skipped. This is a common occurrence in cold wintertime conditions with no local $NH_3$ emission sources, which can result in a high proportion of unattempted retrieval. This can lead to statistical biases when the data are temporally or spatially averaged if not explicitly identified and accounted for in the product. This has been shown in comparisons of CrIS observations with ground-based Fourier transform infrared (FTIR) observations where the lowest concentration values of the CFPR have a positive bias relative to the FTIR observations [20].

Figure 1 is a single-day scene of CrIS surface NH3 pixel observations overlaid on VIIRS true color images for 12 August 2017. This plot contains clouds and non-source regions, as well as various $NH_3$ sources including smoke from large forest fires (including fires in British Columbia), and $NH_3$ agricultural source locations. As non-detects were not accounted for in this figure it does not have values for pixels below the detection limit (i.e., have an absolute $NH_3$ SNR value below 1). An example of this is seen over Hudson Bay in Canada (dark water area near the center of the map) where there are a very limited number of ammonia observations with signals above the CrIS noise level.

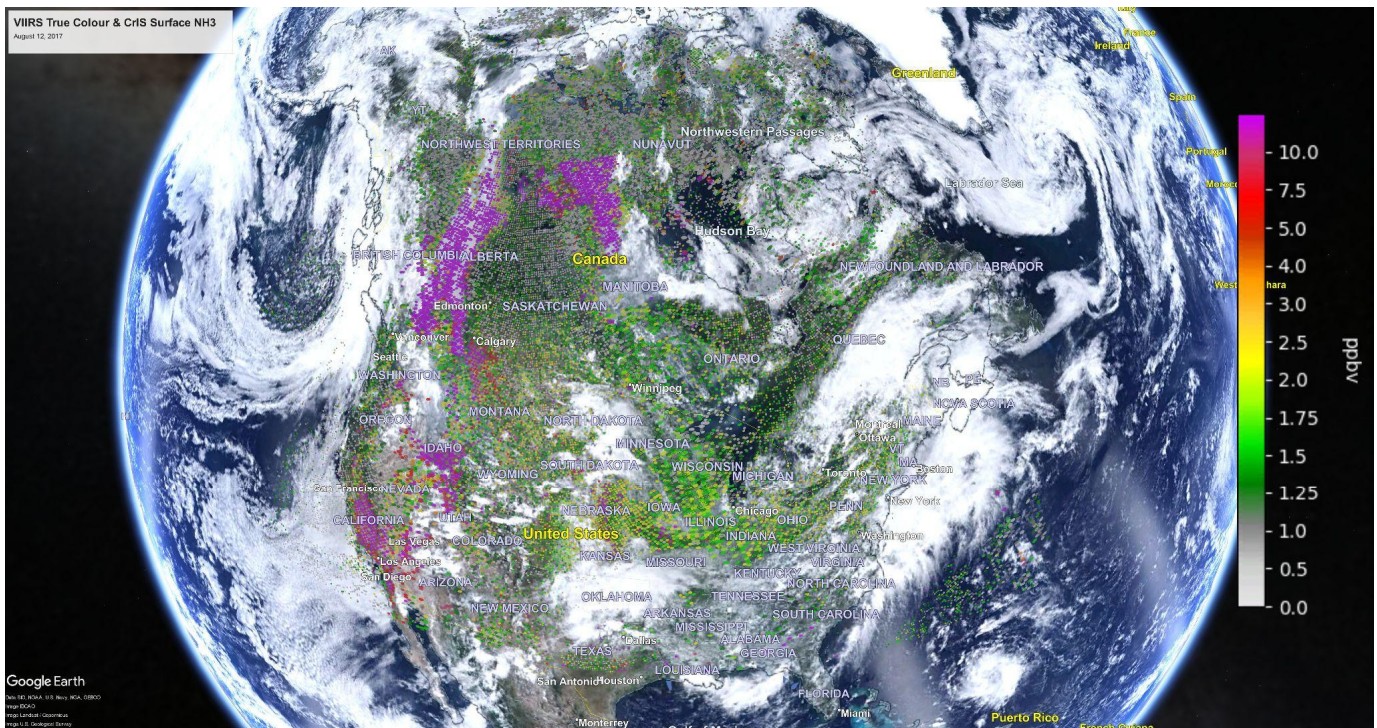

**Figure 1.** CrIS satellite surface $NH_3$ observation pixels combined with VIIRS true color image showing clouds and smoke on 12 August 2017 over Canada and the US. (Underlying VIIRS image obtained from NASA Worldview—https://worldview.earthdata.nasa.gov/, last access: 10 February 2023) (Underlying Google Earth Image obtained from Google Earth© 2022 Google Earth https://earth.google.com/web, last access: 10 February 2023).

### 2.2. VIIRS Observations

The VIIRS instrument is also deployed on the SNPP, NOAA-20, and NOAA-21 satellites. It provides broadband measurements across five infrared channels and eleven visible channels. It is used to generate Earth Data Records (EDR), among them surface temperature, aerosol optical depth, cloud fraction, vegetation indices, and ocean color properties. VIIRS has a much finer spatial resolution than CrIS (750 m vs. 14 km at nadir). Here we use the recently developed University of Wisconsin Space Science and Engineering Center

(SSEC) CrIS IMG product [15] that averages the VIIRS brightness temperatures, reflectances, and cloud fraction over all the VIIRS pixels (~200) contained within the larger CrIS pixels (hereafter referred to as Field of View (FOV)). This product provides additional information that can be used to discriminate between CrIS's FOVs obscured by clouds and FOVs with low $NH_3$ amounts (see Section 3.1). This CIMG product is the basis of the CrIS Ammonia Cloud Detection Algorithm (CACDA) developed to flag cloudy conditions observed by CrIS. The specific elements of the CIMG product used in the CACDA are explained in detail in Appendix A.

### 2.3. In-Situ Surface Observations

In-situ surface observations in non-emission source regions are used to generate representative ammonia surface values for the clear sky observations identified below the $NH_3$ detection limit of the CrIS satellite. Ideally, continuous temporal sampling (e.g., $\leq$1-h intervals) in-situ surface ammonia observations from non-source regions would be used as they can be better matched up with the CrIS overpass time. There are a limited number of continuous surface monitoring observations available in non-source regions, as most surface stations are located in areas to measure elevated $NH_3$ amounts. In this initial study, we used two U.S. Southeastern Aerosol Research and Characterization (SEARCH) network measurement sites in Centreville, Alabama (CTR) (32.90N, 87.25W, rural) and Pensacola, Florida (OLF) (30.55N, 87.38W, suburban) [21,22]. We also used the Pinehouse Lake (PHL) (rural) Canadian Air and Precipitation Monitoring Network (CAPMoN) network site located in northern Saskatchewan, Canada (55.51N, 106.72W). The continuous measurements of $NH_3$ at the PHL site were made using a modified Thermo 42i trace level chemiluminescence based analyzer, which has a detection limit of ~0.1 ppbv [23]. The continuous $NH_3$ observations from these three sites are used to determine representative values based on the different temperature bins as described in Section 3.2.

To demonstrate the impact of accounting for the non-detects in the satellite observations we then compare the instantaneous regional satellite surface ammonia statistics with integrated point source National Atmospheric Deposition Program (NADP) Ammonia Monitoring Network (AMoN) [24], surface station sites that are reported in bi-weekly intervals for the period from May 2012 to May 2021 (see Section 4). The AMoN stations use Radiello diffusive passive samplers for surface $NH_3$ measurements, which have a detection limit of ~0.1 ppbv [25]. An interactive map of station location and metadata information is provided at https://nadp.slh.wisc.edu/maps-data/amon-interactive-map/ (accessed on 10 February 2023)

## 3. Identifying and Accounting for Non-Detects

Two of the challenges in accounting for non-detects in satellite retrievals are identifying "clear" non-detects in a sample, and assigning them reasonable representative values. Often retrievals either ignore these conditions, or do not identify them at all and implicitly assign them a value (e.g., zero or an a-priori profile value). Here we present a procedure that both explicitly identifies these non-detect observations and assigns them a representative value based on well-calibrated surface observations. This procedure differentiates non ammonia spectral signals in clear and cloudy conditions, and then accounts for these observations in clear conditions by inserting representative ammonia values derived from in-situ observations in background regions.

### 3.1. Cloud and Non-Detect Flag (CNF)

The CACDA was developed to separate the low signal FOVs due to clouds severely attenuating the signal, which provides no information on $NH_3$ amounts, from those due to ammonia amounts below the detection limit of the sensor. Appendix A contains a detailed description of the development of the CACDA. The CACDA is specifically designed to be used for ammonia and incorporates the VIIRS visible and infrared cloud information from the CIMG product as well as the VIIRS cloud fraction. This cloud information is then

used to develop the CFPR Cloud and Non-detect Flag (CNF) shown in Table 1. The CNF is generated for each CrIS FOV except for those where the retrieval did not converge. Note that some FOVs with NH$_3$ values above their detection limit will be flagged as cloudy (CNF = 1). This implies that the NH$_3$ signal is strong enough to be detected through thin clouds, but the uncertainty in the retrieved NH$_3$ will be greater than for a cloud-free FOV.

**Table 1.** Table containing Cloud and Non-detect Flag (CNF) categories found in the CFPR Level 2 NH3 product dataset, as well as the physical conditions the CNF categories represent.

| Cloud Flag | Descriptor | Comment |
|:---:|:---:|:---:|
| −1 | No cloud information | Corresponding VIIRS cloud information was missing. |
| 0 | Clear retrieval | Retrieval under cloud-free conditions. |
| 1 | Cloudy retrieval | Retrieval under cloudy conditions |
| 2 | Smoke cloud | CrIS FOV that were initially identified by VIIRS as cloudy, but were identified as smoke plumes by the CACDA. |
| 3 | Non-detect | Representative data for cloud-free CrIS FOV below the detection limit of the sensor. |

Note: Cloudy pixels with no signal are not retrieved or included in the dataset. "Cloud Flag" is the variable name used in the product files for this flag.

Implementing the CACDA as a post processing step has the advantage of reducing the computation burden of performing retrievals. The post-processing implementation in combination with the CFPRs skipping retrievals below the detection limit, reduces the computational burden of global retrievals by 36% on average, while still accounting for the roughly 12% of pixels that are non-detects in the CrIS ammonia dataset.

The CNF allows the user of the CFPR Level 2 product to select which data to use in their analyses. Note that if the FOV is flagged as cloudy and there was no retrieval because the NH$_3$ signal is low, this FOV is excluded from the dataset. It is recommended for most analyses that the user select the CNF flag that keeps all FOVs except those with CNF = 1.

Figure 2 shows the CNF for the same CrIS data used in Figure 1. Over the non-source regions (e.g., Hudson Bay in Canada) there are a number of non-detect (CNF = 3; purple) FOVs added to the image. Thick smoke plumes are also identified by the CACDA algorithm (CNF = 2; red). Thick cloudy regions are white without any retrieved values as the instrument cannot detect ammonia below the thick clouds. There are some locations (mainly near the edges of cloud systems) where the CACDA algorithm detects thin clouds and a retrieval was performed (CNF = 1; yellow); these retrievals have higher uncertainties and should be used with caution.

*3.2. Non-Detects Values*

In-situ data was used to generate representative ammonia concentrations under CrIS non-detect conditions. As noted earlier, there are a very limited number of continuous surface stations in non-source locations. Here, we used continuous measurement of surface NH$_3$ and temperature data, from three background measurement sites. Two stations are from the SEARCH measurement network (CTR and OLF) for the period of 2012–2016, and one is from CAPMoN measurement network (PHL)) from 2016–2017. We used the continuous in situ measurements between 13:00–14:00 h to match the in situ measurement to the satellite overpass (i.e., 13:30 h) on a daily basis. As temperature is one of the main drivers in the volatilization of ammonia from the surface to the air [26,27] we generated these initial representative non-detect ammonia concentration values as a simple function of temperature. As more data becomes available this approach can be refined to include other potential dependences (i.e., wind speed, humidity, soil moisture, etc.).

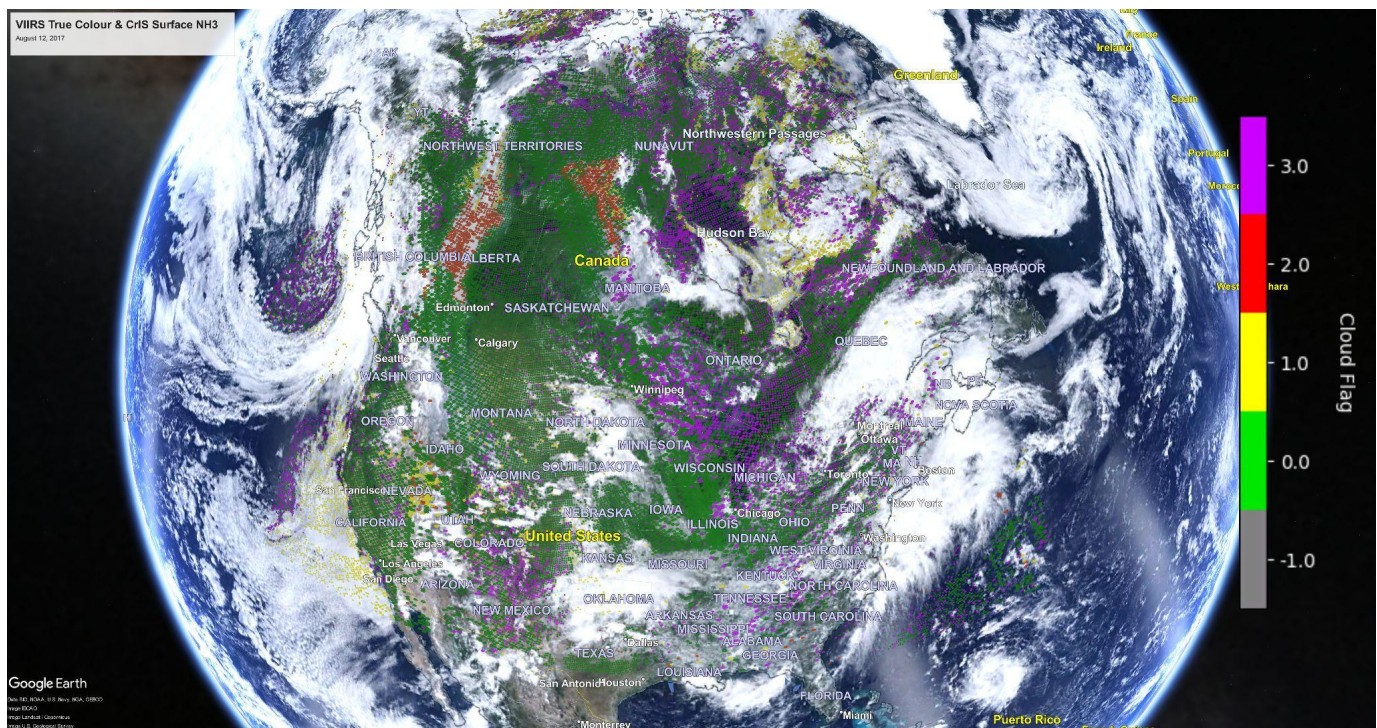

**Figure 2.** The CFPR cloud and non-detect flag for the same 12 August 2017 image shown in Figure 1. (Underlying VIIRS image obtained from NASA Worldview-https://worldview.earthdata.nasa.gov/, last access: 10 February 2023) (Underlying Google Earth Image obtained from Google Earth© 2023 Google Earth https://earth.google.com/web, last access: 10 February 2023).

The daily matched values are then used to calculate the median values of surface $NH_3$ concentrations for different surface temperature bins at each of the three stations. More than 50 observations were used to calculate the median value for each of the temperature bins. Then, to account for the geo-meteorological variability in North America the average value of the median surface $NH_3$ concentration from all three stations is calculated and used for the representative non-detect values in Table 2.

**Table 2.** Table of non-detect $NH_3$ surface values as a function of surface temperature.

| Temperature (°C) | Non-Detect $NH_3$ Surface Values (ppbv) |
|---|---|
| <−25 | 0.0 |
| [−25 to −20] | 0.0423 |
| [−20 to −15] | 0.0732 |
| [−15 to −10] | 0.0959 |
| [−10 to −5] | 0.1705 |
| [−5 to 0] | 0.1720 |
| [0 to 5] | 0.2244 |
| [5 to 10] | 0.2666 |
| [10 to 15] | 0.3863 |
| >15 | 0.4649 |

Figure 3 is the same surface ammonia example used in Figure 1, but with the non-detects shown in Figure 2. The clear-sky non-source conditions with sparse retrieved values (e.g., Hudson Bay, Arctic) now include the representative background values that will help provide better statistics for gridded and averaged products.

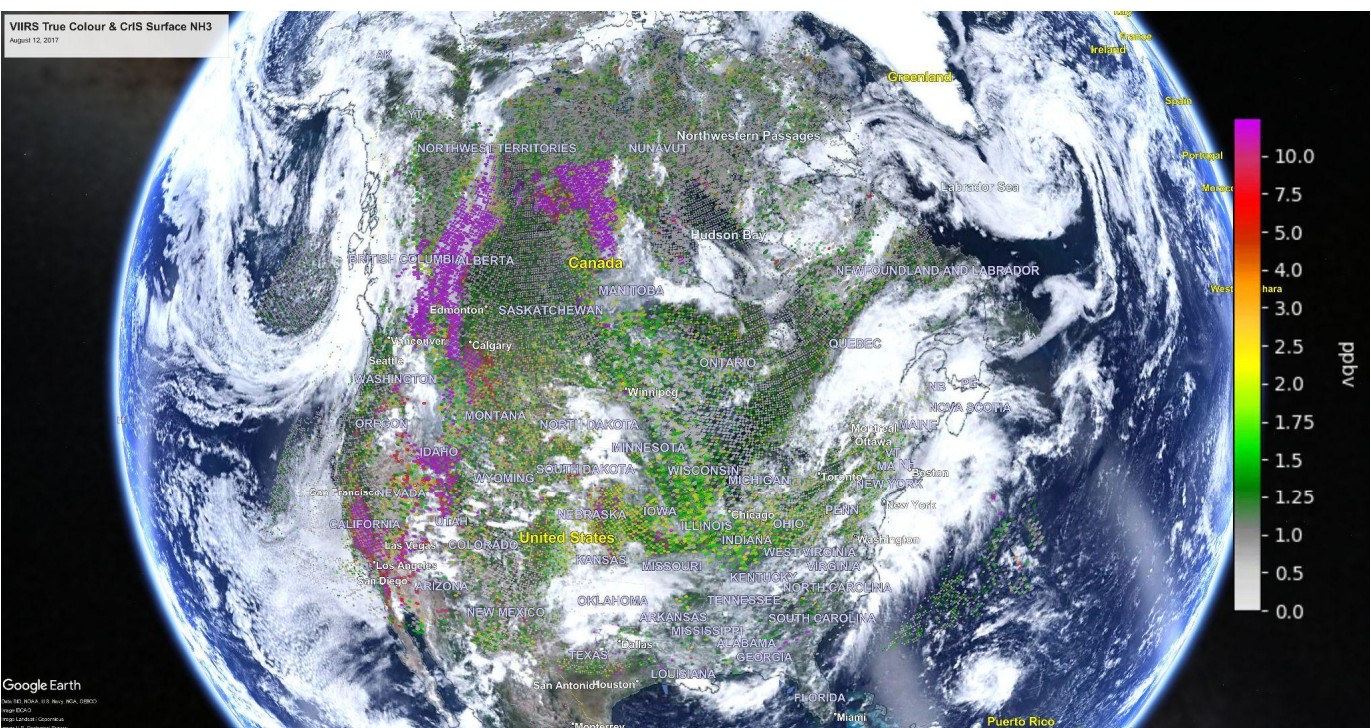

**Figure 3.** Accounting for non-detects in the 12 August 2017 daily plot of CrIS satellite surface NH$_3$ observations shown in Figure 1. (Underlying VIIRS image obtained from NASA Worldview— https://worldview.earthdata.nasa.gov/, last access: 10 February 2023) (Underlying Google Earth Image obtained from Google Earth© 2023 Google Earth https://earth.google.com/web, last access: 10 February 2023).

Figure 4 presents three overlaid histograms illustrating the changes in CrIS NH$_3$ distribution over the PHL surface site when non-detects are included, and how they compare to the in-situ distribution. PHL is a background surface location that experiences a broad range of surface temperatures from winter to summer, and is a good station to demonstrate the procedure of accounting for non-detects. The in-situ values in the histograms were chosen by co-locating the PHL average surface observations with CrIS observations, including the non-detects (blue), that occurred in the same hour (i.e., 13:00–14:00 h) as CrIS overpass time (i.e., ~13:30 local time) and 15 km of the center of a CrIS footprint. For values > 0.5 ppbv the satellite histogram values with (blue) and without (red) non-detects are the same, which is expected as the additional non-detect values have a maximum value below 0.5 ppbv (see Table 2). Note that for the instantaneous observations below 0.5 ppbv adding the non-detects significantly improves the agreements with the in-situ data. This is expected given that the representative data used for the non-detects was generated using the in-situ data from the PHL site. This result simply demonstrates that the methodology implemented works as designed, and that accounting for non-detects will improve statistically averaged background conditions. This is also shown in the mean and median values: accounting for non-detects reduces the overall 2-year mean (median) values of the datasets from 1.3 (1.1) ppbv to 0.8 (0.5) ppbv, approaching the in-situ mean values of 0.5 (0.4) ppbv. Also provided are the median absolute deviations (MAD) values of the distributions, where half the data points are within one MAD of the median value in either direction. An evaluation of including non-detects with bi-weekly AMoN observations is provided in Section 4.

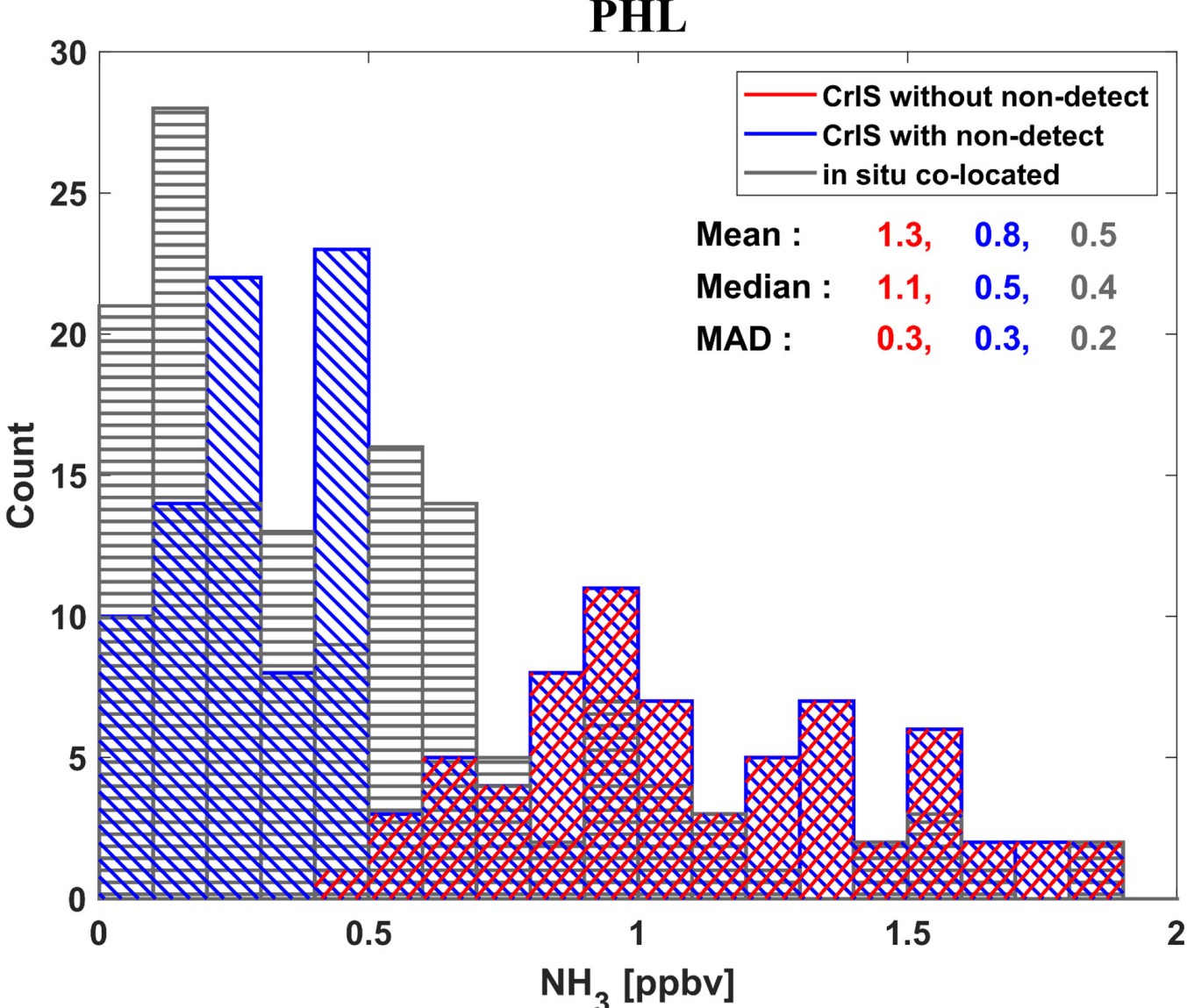

**Figure 4.** Histograms demonstrating the effect of including non-detects over a PHL station from 2016 to 2017 for PHL. The CrIS histograms are shown both without non-detects (red) and the overplotted accounting for non-detects (blue); the in-situ surface observations for the PHL surface station are shown in black.

## 4. Surface Evaluations including Non-Detects

Provided here are comparisons of the distribution of CrIS observations with and without non-detects against in-situ surface observations. The purpose here is not a detailed validation of the CrIS satellite observations, especially since sampling differences between the in-situ point source surface stations and coarser CrIS observations, which can have a significant impact in inhomogeneous conditions, have not been taken into account. The main purpose here is to demonstrate the impact of including non-detects on the CrIS distributions, and how these new distributions compare to an expected distribution over a region as observed by in-situ surface observations. It is important to note that the AMoN observations are bi-weekly integrated observations, whereas the corresponding CrIS values are 2-week averages of instantaneous 01:30 daytime overpass observations. For this evaluation, we selected background AMoN surface stations, with a significant number of very low (<1.0 ppbv) ambient ammonia values. The impact of accounting for non-detects is expected to be more significant at such stations. We chose stations where the impact of

spatial sampling differences between in-situ point source surface stations and coarser CrIS observations is not large (i.e., a more homogenous ammonia field). This gives a reasonable comparison to CrIS observations. We used CrIS Level 2 $NH_3$ observations with quality flag $\geq 4$, cloud flag $\neq 1$, co-located within 15 km of the AMoN station. Figure 5 shows that accounting for non-detects shifts the distribution towards smaller values that are in closer agreement with the in-situ observations (black).

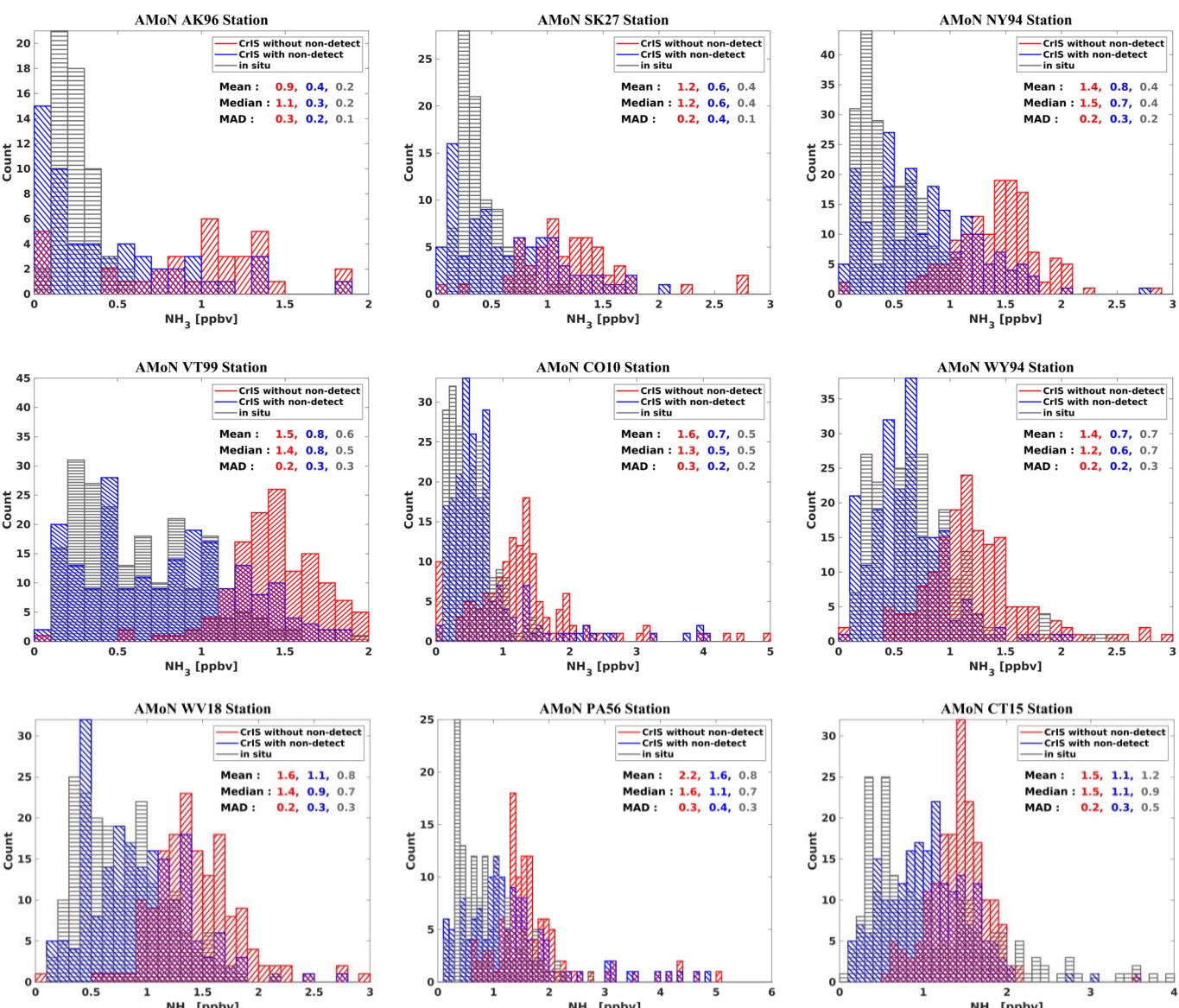

**Figure 5.** Histogram of bi-weekly average $NH_3$ concentrations from in situ and CrIS satellite observations with and without non-detect values for AMoN stations located in background regions.

## 5. Application to CrIS NH₃ Satellite Observations

Non-detects are accounted for at the single pixel level (Level 2), since the conditions that lead to a non-detect vary from pixel-to-pixel. However, the overall impact of accounting for non-detects is more evident when averaging the single pixel values (Level 2) to generate gridded (Level 3) data. CrIS gridded and averaged Level 3 values were created on a uniform $0.1° \times 0.1°$ grid with quality flag $\geq 4$ [17] and averaged over the 9-year time period from May 2012 to May 2021 (Figure 6). The top left plot in Figure 6 shows the multi-year average plot not accounting for non-detect pixels, whereas the top right shows the corresponding plot with non-detects included. Comparing these two plots shows

that in general, accounting for non-detects reduces the average surface values in non-source or background regions. At the same time, the values in the regions with strong year-round sources remain relatively unchanged. Differencing the two datasets (Figure 6, lower left) shows that the largest impacts are in the regions with no significant sources, or predominantly seasonal sources such as fires. These are also the regions with the greatest number of non-detects (Figure 6, lower right) since the ammonia values are frequently below the detection limit of the satellite sensor. The corresponding results for the total column amounts are provided in Appendix C.

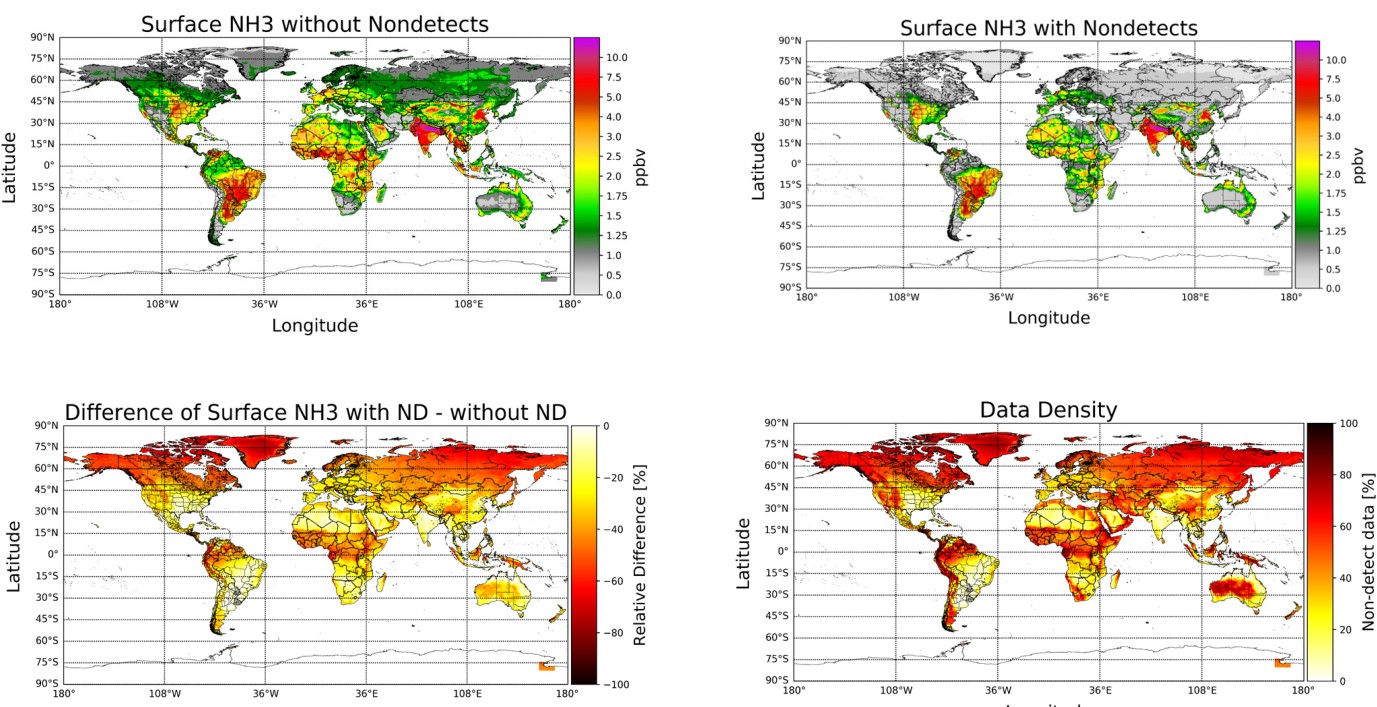

**Figure 6.** (**Upper left**) CrIS global averaged surface observations for the May 2012 to May 2021 period without including non-detects (ND); (**Upper Right**) CrIS observations over the same period, but including the non-detects; (**Lower Left**) the relative difference between including and excluding non-detects; (**Lower Right**) percentage of non-detects added to the global dataset.

The impact of including the non-detects can be quantified by looking at the relative differences as a function of the ammonia amounts, as in Figure 7. The average surface values decrease by over 50% in non-source regions (<1 ppbv), but by less than 5% in strong source regions (>7.5 ppbv). A similar impact is observed when looking at the fraction of non-detects as a function of ammonia amounts (Figure 8). The percentage of non-detects ranges from ~70% in low source background regions (<1 ppb) to <5% in strong source regions (>7.5 ppbv). Over this period of time, no regions experience an increase in the average measured ammonia level as a result of the inclusion of non-detects. These show that the impact of including the non-detects for total column mean values is similar to the surface values, as would be expected, since the total column values are the integrated values of the profile values generated from the surface values (see Appendix B).

As indicated earlier, seasonal variability can also influence the impact and occurrence of non-detects. Seasonally binned multi-year annual average $NH_3$ values (Figure 9) and percentage of non-detects (Figure 10) show that regions with strong seasonal variability will have more non-detects in cooler weather. This is due to a combination of factors: $NH_3$ sources are much weaker (e.g., there is no fertilizer application in winter); $NH_3$ emissions are much lower at colder temperatures, and biomass burning, another strong but localized $NH_3$ source, is also rare in cold weather. This is evident in many regions

around the globe. For example, NH$_3$ source regions above 45 N latitude in North America and Europe have significantly more non-detects in the winter months (upper left) than during warmer periods (lower left), when fires can be significant localized sources. In some of these regions, the percentage of non-detects can range from >60% in the wintertime to <15% in the warm seasons.

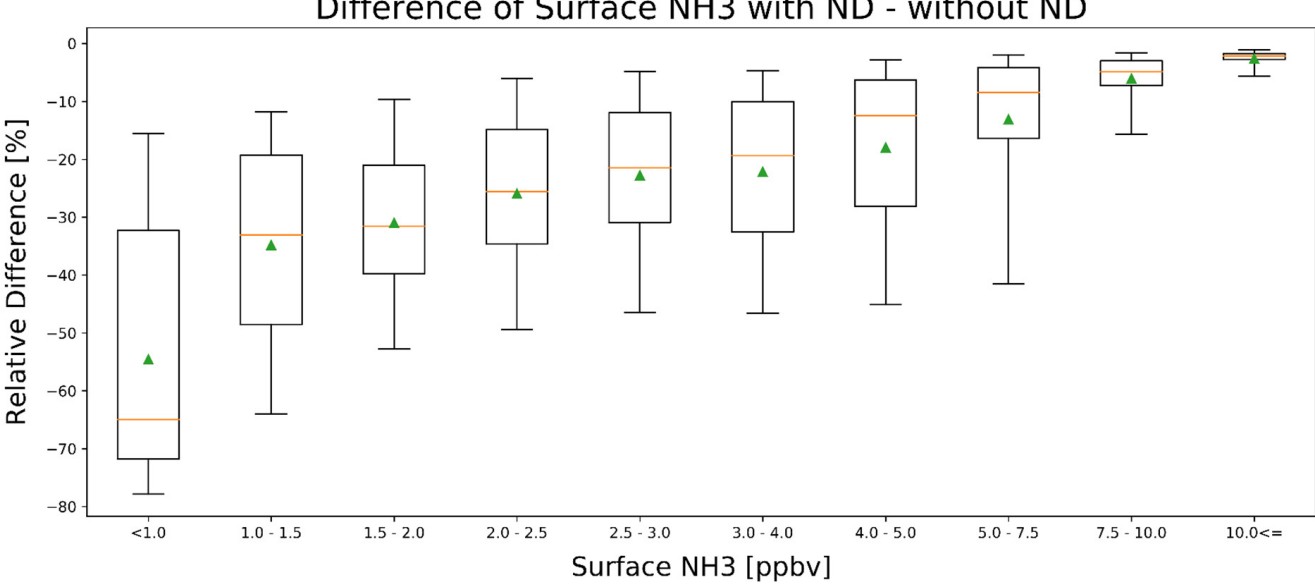

**Figure 7.** Box-and-whiskers plot of the relative differences in the annual gridded surface NH$_3$ values from including non-detects as a function of surface value magnitudes. Data points use the same spatial and temporal period as Figure 6. The whiskers are the 5th and 95th percentiles, the box is the 25th and 75th percentile, where the orange line is the median value, and the triangle is the mean value in each bin. Data points are binned using the original surface NH$_3$ values without non-detects.

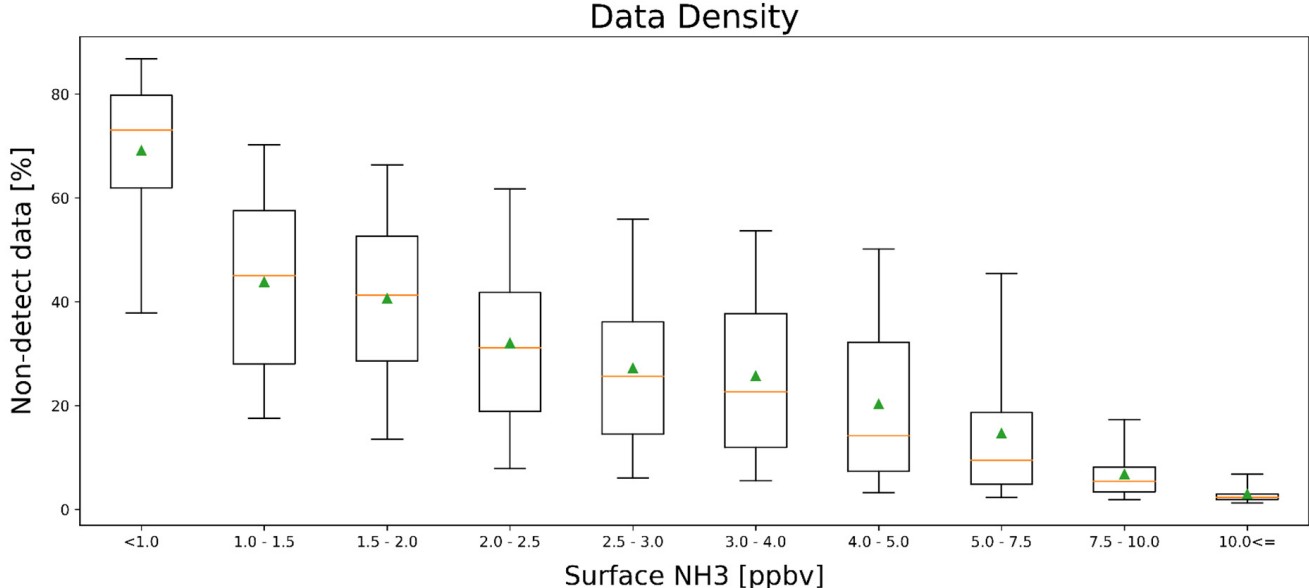

**Figure 8.** The percentage of non-detect values as a function of surface ammonia concentrations, for the same time period as Figure 7.

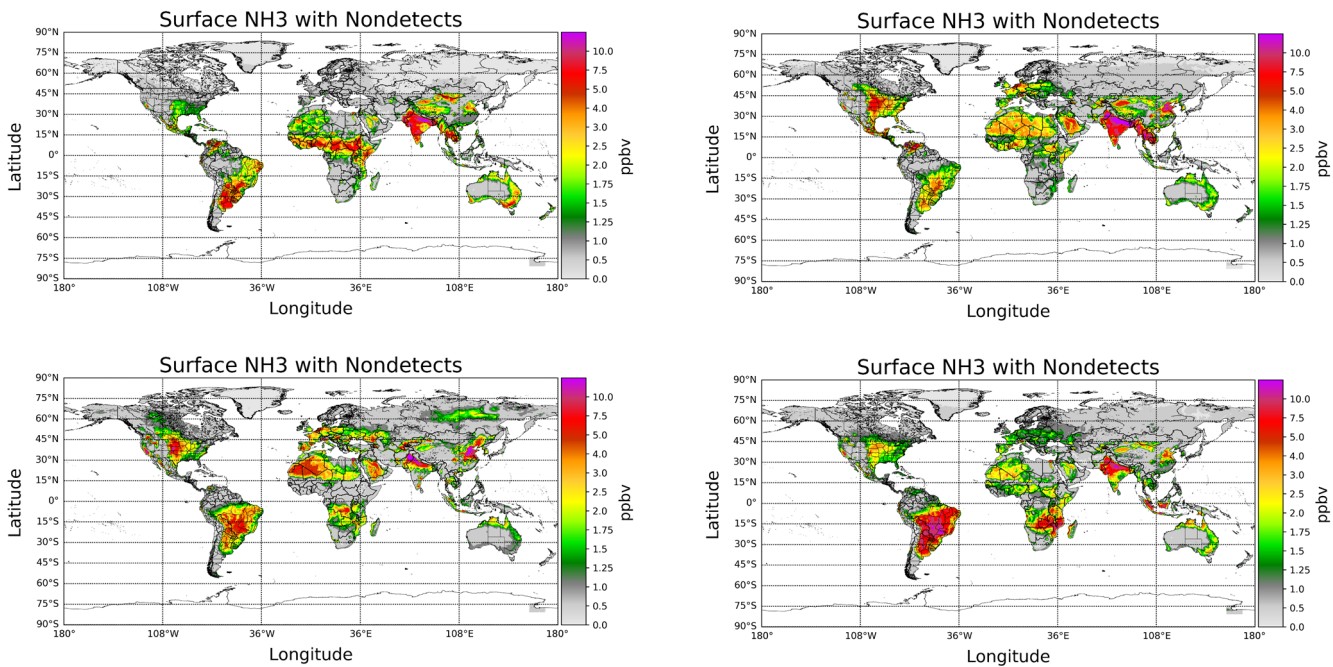

**Figure 9.** CrIS multi-year (May 2012–May 2021) surface observations split up into the seasons of (**Upper Left**) winter (December-January-February), (**Upper Right**) spring (March-April-May), (**Lower Left**) summer (June-July-August), and (**Lower Right**) fall (September-October-November).

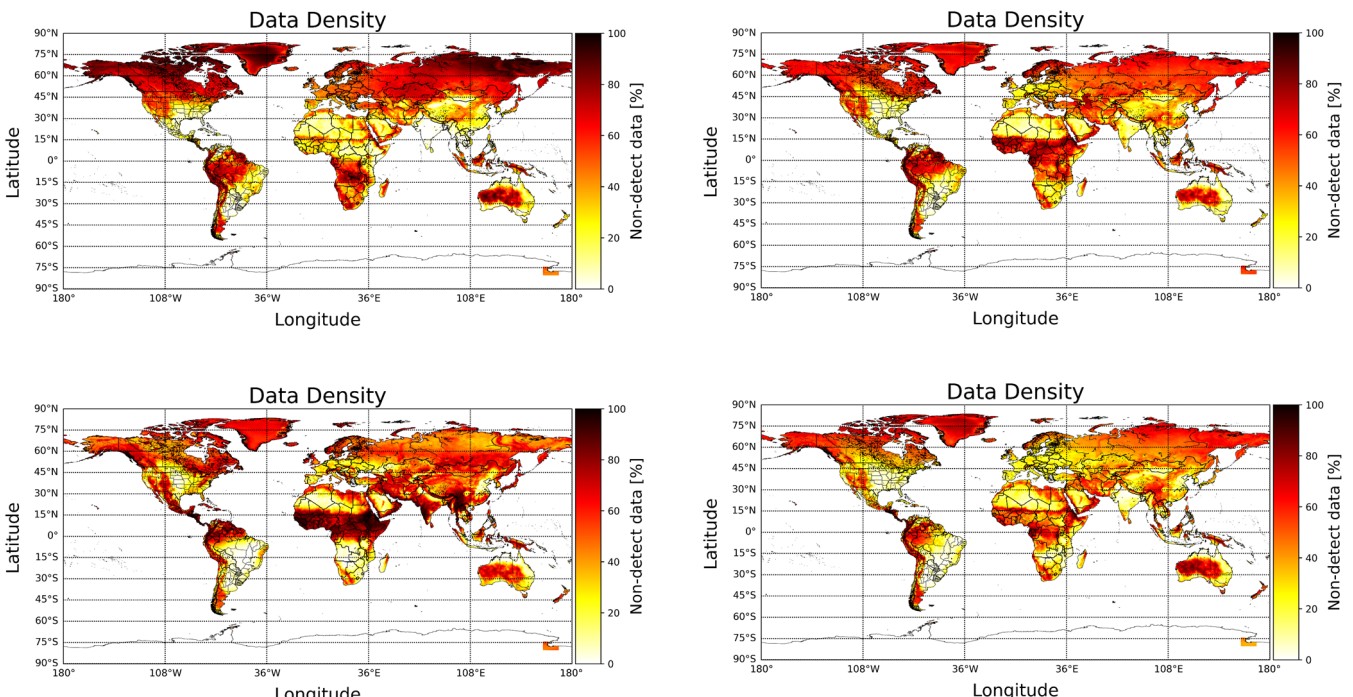

**Figure 10.** CrIS multi-year (May 2012–May 2021) percentage of non-detect values split up into the seasons. (**Upper Left**) winter (December-January-February), (**Upper Right**) spring (March-April-May), (**Lower Left**) summer (June-July-August), and (**Lower Right**) fall (September-October-November).

## 6. Conclusions

Even though the CrIS sensor has an excellent sensor signal-to-noise ratio (~1600) in the spectral region used for $NH_3$ retrievals, accounting for non-detects can have an impact on satellite $NH_3$ products. The impact is most evident in regions or seasons with weak

emissions. In this study, we developed a post-retrieval processing approach to identify and take into account non-detects in the CFPR $NH_3$ single observation retrievals. This involved creating a CFPR Cloud and Non-detect Flag (CNF), which required the development of a new CrIS Ammonia Cloud Detection Algorithm (CACDA) and the generation of non-detect distribution of surface concentrations. The CACDA separates the CrIS FOVs with low ammonia signals due to clouds from those due to low amounts of ammonia below the detection limit of the sensor. The CrIS FOVs that are placed in the latter group are then populated with representative data values. The frequency of non-detects ranged from <5% in strong source regions, to >~70% in background regions (<1 ppbv); in regions with strong seasonal sources such as fires this frequency varies seasonally. The resulting relative impact of accounting for non-detects on the ammonia surface concentrations goes from decreasing values by over 50% in non-source conditions (<1 ppbv), to less than 5% in strong source regions (>7.5 ppbv). Comparison with in-situ surface stations shows satellite $NH_3$ observations better represent the lower distribution of surface stations when non-detects are included, even given the potential difference in sampling between the two types of observations. An additional benefit of identifying the non-signal conditions and handling the non-detects in a post-processing step is that it greatly reduces the computational burden of global retrievals as this removes the need for performing full retrievals when there is no $NH_3$ signal in the satellite spectrum. For example, this approach of adding in non-detects in a post-processing step for daytime retrievals reduces the computation burden by 36% on average, while still allowing for the roughly 12% of pixels that are non-detects to be accounted for. In the future, as more instantaneous background surface observations over various background conditions become available, they can be used to refine the inserted representative data used to populate the satellite non-detects. Non-detects are available in the CFPR product starting from Version 1.6. Version 1.6 has already been used to obtain improved emission estimates of $NH_3$ over the United Kingdom and globally, respectively [28,29].

**Author Contributions:** Data curation, D.T., G.Q., J.O. and J.B.; Formal analysis, M.W.S., K.E.C.-P., S.K.K., S.F., E.D., E.C., N.T., D.T., G.Q., J.O. and J.B.; Investigation, E.W.; Methodology, E.W., M.W.S., K.E.C.-P. and S.K.K.; Software, E.W.; Visualization, S.F., E.D., E.C. and N.T.; Writing—original draft, E.W., M.W.S., K.E.C.-P. and S.K.K.; Writing—review and editing, E.W., M.W.S., K.E.C.-P., S.K.K., S.F., E.D., E.C., N.T., D.T., G.Q., J.O. and J.B. All authors have read and agreed to the published version of the manuscript.

**Funding:** The Research done by Karen Cady-Pereira on this project at AER was funded by NASA grant number 80NSSC18K156 and by NASA grant number 80NSSC21K1963. The CIMG product from the University of Wisconsin was funded by NASA Goddard Space Flight Center (GSFC) grant number 80NSSC22K0713.

**Data Availability Statement:** The Level 3 version of the CrIS CFPR Level 2 ammonia data created by Environment and Climate Change Canada data [17] used in this study is openly available at: https://hpfx.collab.science.gc.ca/~mas001/satellite_ext/cris (last access: 10 February 2023). The use of this data is subject to the Open Government License—Canada (https://open.canada.ca/en/open-government-licence-canada, last access: 10 February 2023). The Python/Matlab code used to create any of the figures is available upon request to mark.shephard@ec.gc.ca. The University of Wisconsin Space Science and Engineering Center (SSEC) CIMG product is publicly available at NASA GES DISC https://disc.gsfc.nasa.gov/datasets/SNDRSNCrISL1BIMG_2/summary (last access: 10 February 2023). The AMoN data is publicly available from NADP at https://nadp.slh.wisc.edu (last access: 10 February 2023).

**Conflicts of Interest:** The authors declare that they have no conflict of interest.

## Appendix A  CrIS Ammonia Cloud Detection Algorithm (CACDA)

The CIMG files contain VIIRS data averaged over all the VIIRS pixels within each CrIS pixel, hereafter referred to as an FOV for clarity. Among the variables in the CIMG files are cloud fraction, and brightness temperatures (BT) in the infrared channels. Separate

averages over the clear and cloudy VIIRS pixels are also provided (Figure A2, top right and lower left; blank regions in either plot indicate CrIS FOVs that are completely cloudy, or completely clear).

An initial attempt to classify all CrIS FOVs with a cloud fraction greater than 0.2 as cloudy rejected a significant fraction of FOVs that provided low but physically reasonable $NH_3$ amounts, as shown in the left panel of Figure A4. Instead, we utilized the brightness temperature (BT) data in the 10.8 μm VIIRS infrared channel, which encompasses the spectral feature used by the CFPR algorithm for the $NH_3$ retrievals, to create a more subtle mask. Note that for demonstration purposes retrievals in Figures A2 and A4 were carried out over all FOVs.

We hypothesized that if the brightness temperature difference between the clear and cloudy portion of an FOV was below some threshold in the band where $NH_3$ is radiatively active, then the cloudy area in that FOV would still allow a reasonable fraction of the $NH_3$ signal to reach the CrIS sensor. To determine this threshold we calculated the BT at the top of a cloudy layer as a function of cloud optical depth, using Beer's Law. We did this for a range of surface skin temperatures, then subtracted the calculated BT from the surface BT. As stated in Section 2, the $NH_3$ spectral feature is in a window region, where the BT measured at the top of the atmosphere is very close to the BT measured at the surface if there are no clouds (Figure A3). We found that a difference of 25 K or less allowed 45% to 70% of the $NH_3$ signal to pass through the cloud. Accepting FOVs that pass this test leads to significantly fewer rejections, as can be seen by comparing the left and right panels in Figure A4. It could be argued that this lower criterion is simplistic and too loose and allows for cloud-contaminated data to be accepted. However, our objective is to include as much information on $NH_3$ amounts as possible and let the user decide what they want to include in their analysis. This is in effect a quality control test, and such tests frequently require empirical cutoffs. While the presence of some very thin clouds may interfere with ammonia retrievals, they do not totally obscure the $NH_3$ signal and thus provide some information. These are identified in the CFPR cloud and non-detect flag (CNF = 1) allowing the user the flexibility to determine if they would like to use these less reliable "thin cloud" pixels in their analysis. Below are the detailed steps of the CACDA to generate the CNF (also represented as a flowchart in Figure A1):

1. If the CIMG cloud fraction in an FOV is less than 0.25, the FOV is flagged as clear (CNF = 0).
2. If the CIMG cloud fraction is greater than 0.9 the FOV is flagged as cloudy. If a retrieval was performed then it will have (CNF = 1), otherwise, the pixel will be skipped from the CFPR product as we currently do not consider non-signal cloudy pixels.
3. If the CIMG cloud fraction is greater than or equal to 0.25 and less than or equal to 0.9, but the BT difference between the clear and cloudy averages is less than 25 K, the FOV is flagged as clear; otherwise, it is flagged as cloudy (CNF = 1).
4. If a pixel is identified in the first three steps using CIMG as being clear, but no retrieval was attempted($NH_3$ SNR < 1 in the $NH_3$ window) then the pixel is flagged as a Non-detect(CNF = 3).
5. If a pixel is identified in the first three steps using CIMG as being cloudy, but the estimated $NH_3$ SNR > 5 in the $NH_3$ retrieval window, then the CrIS FOV is flagged as smoke filled (CNF = 2). This final step is required because the VIIRS processing flags often incorrectly flags thick smoke pixels as clouds. Note that the $NH_3$ SNR is a linear function of the ammonia spectral signal divided by the CrIS noise in the $NH_3$ spectral region [16], which provides a measure of the spectral strength of the $NH_3$ signal. A high $NH_3$ signal over a cloudy FOV is a strong indicator of the presence of $NH_3$ from fires. Thus, applying this last step on cloudy pixels will retain these smoky pixels, which are a source of large $NH_3$ concentrations. An example of this is shown in Figure 2 where thick smoke pixels from large forest fires are shown in red.

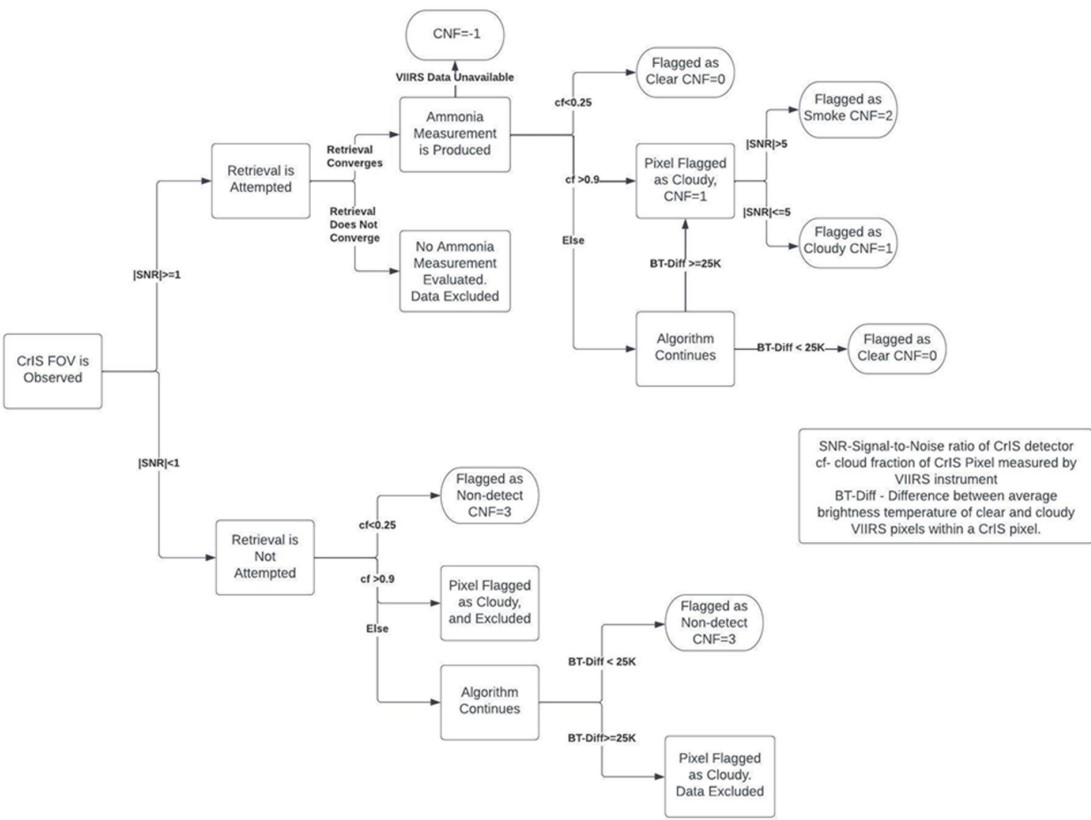

**Figure A1.** A flow chart describing the CrIS Ammonia and Cloud Detection Algorithm (CACDA).

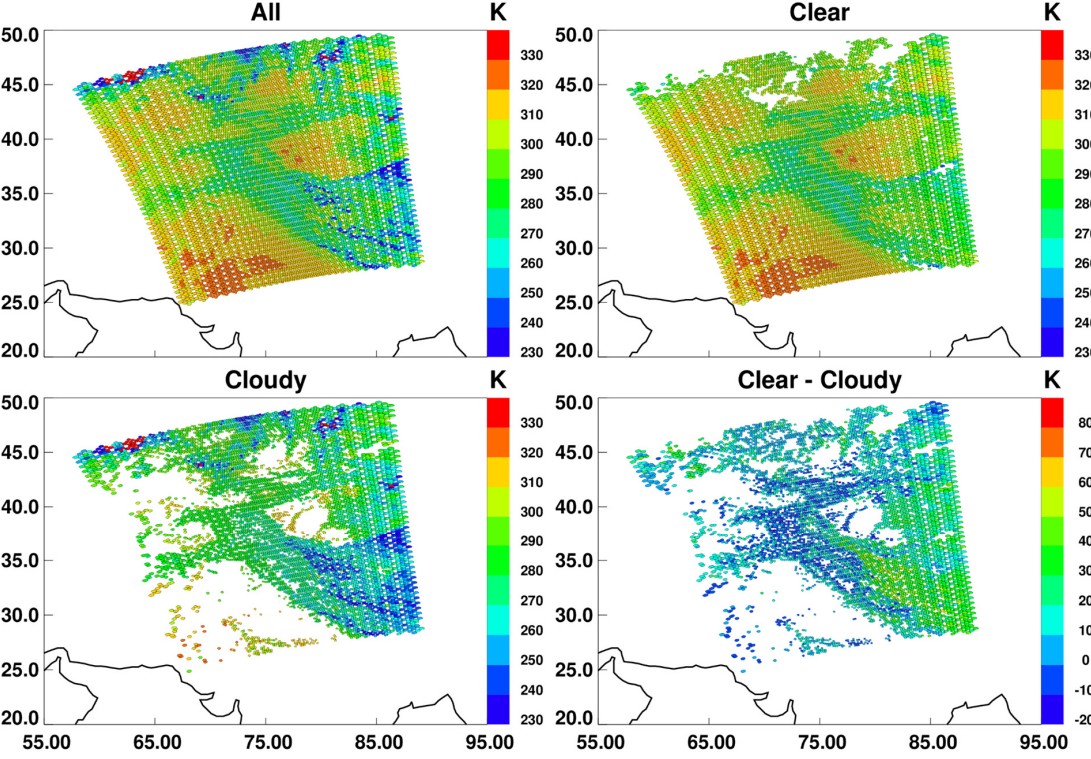

**Figure A2.** Plots of IMG 10.8 μm channel brightness temperatures. (**Upper Left**) average of all VIIRS pixels in each CrIS FOV. (**Upper Right**) average of clear pixels. (**Lower Left**) average over cloudy pixels. (**Lower Right**) difference between clear and cloudy average BTs.

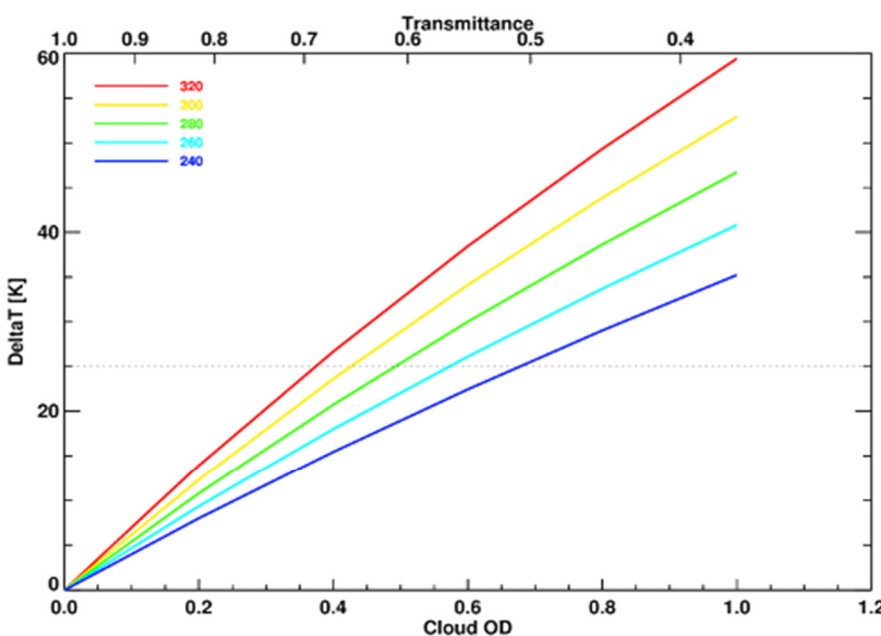

**Figure A3.** Difference in brightness temperature between the clear and cloudy portions of FOVs as a function of cloud OD (or transmittance) for a range of surface temperatures.

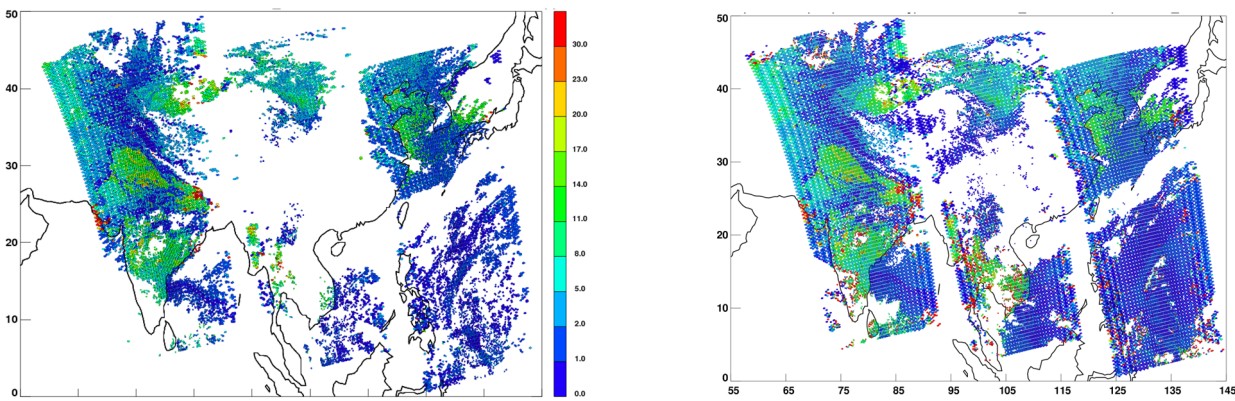

**Figure A4.** (**Left**) Surface CrIS NH$_3$ from 22 April 2015 with only CrIS FOVs with cloud fraction less than 0.2. (**Right**) Same data with only FOVs determined to be clear as described in Section 3.1.

## Appendix B  Non-Detect CFPR Product Parameters

It is desirable that the identified non-detect pixels also contain all the additional parameters that are provided in the CFPR product (e.g., averaging kernels, error covariances, etc.) for the retrieved pixels. Thus, representative values are provided in the files to go along with the non-detect surface ammonia concentration values. The representative averaging kernel, measurement covariance error matrix, and total covariance error matrix were derived by averaging CrIS retrievals with very little signal ($|$NH$_3$ SNR$|$ < 1.01) close to the sensor's detection limit in a region over North America 100 to 80 W and 45 to 60 N. These average values for 2017 binned by season are shown in Figure A5. For CFPR v1_6 we selected just the average winter values as these had the lowest signals. An NH$_3$ profile is generated by scaling an unpolluted profile [14] by the representative non-detect surface value, with a corresponding total column value computed by integrating up the profile NH$_3$ concentrations.

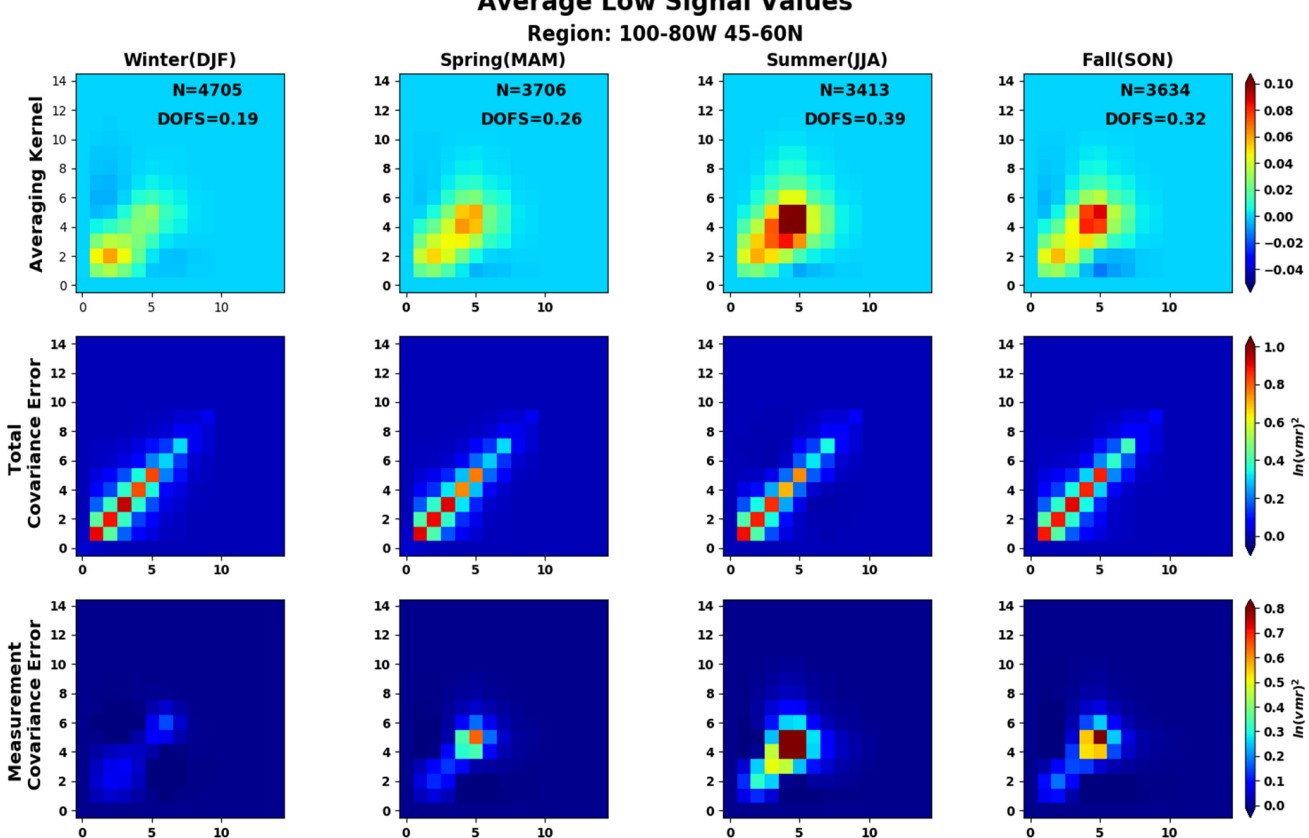

**Figure A5.** Seasonal plots of the averaging kernel, measurement, and total error covariances matrices plotted as a function of the retrievals levels starting from the surface (level 0) up to the top-of-the atmosphere (level 14) from retrievals with low signals ($NH_3$ SNR < 1.01) in 2017 for a region over central Canada.

## Appendix C  Total Columns

The retrieved profiles can be integrated to provide vertical column densities. Thus, the corresponding total column values associated with the annual averaged values shown in Figure 6 are provided in Figure A6. Similar to Figure 6, the identification and accounting for non-detects has a greater impact in the non-local source regions where there are more occurrences of conditions below the detection limit of the sensor (Figure 6 bottom right).

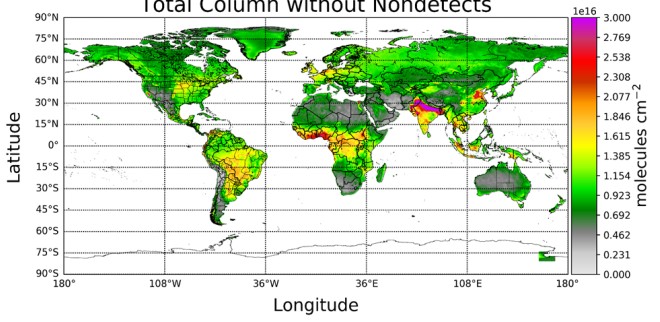
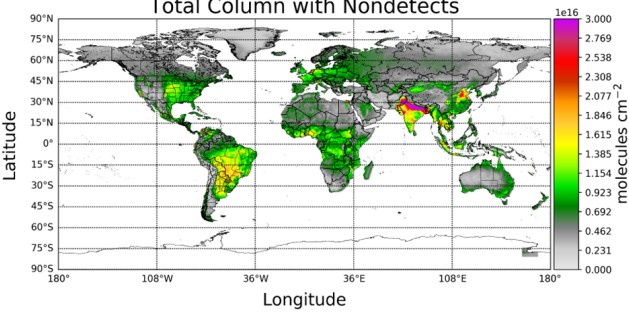

**Figure A6.** *Cont*.

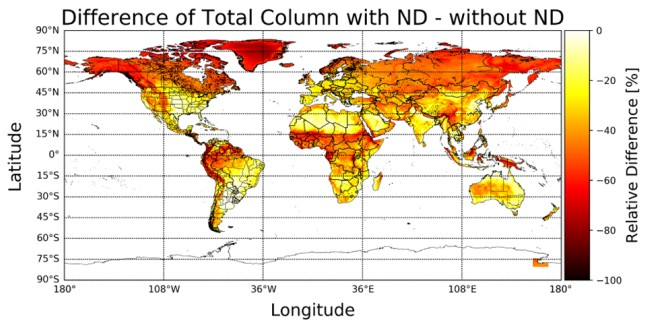
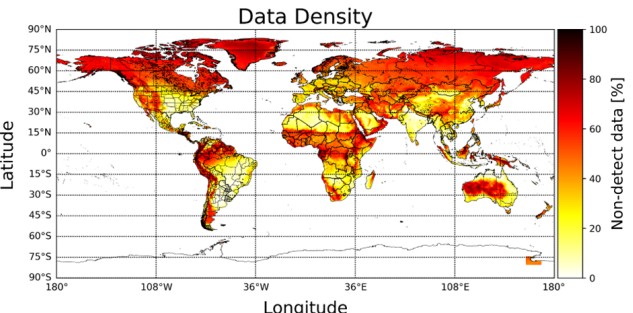

**Figure A6.** (**Upper Left**) The CrIS global averaged vertical total column values for May 2012 to May 2021 period without including non-detects. (**Upper Right**) The CrIS observations over the same period, but including the non-detects. (**Lower Left**) The relative difference between including and excluding non-detects (Minimum and maximum values for normalization are set to −100 and 0 respectively). (**Lower Right**) The percentage of non-detects added to the global dataset is the same for both the surface concentrations and total column, (just provided for reference).

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
