# Peer review of "Accounting for Non-Detects: Application to Satellite Ammonia Observations"

_remotesensing, doi:10.3390/rs15102610_

Round 1

Reviewer 1 Report

This paper proposes a method to estimate the ammonia concentrations in the FOV with lower signals than the detection limit of the CrIS NH3 product. The paper is generally well-written. The topic fits well with the aim of the journal. On the other hand, some unclear points should be pointed out. I recommend the paper be published after addressing the following comments.

Major comments:

The authors should precisely describe the background, necessity, and importance of this study in Section 1.

e.g., How does the non-detect measurement affect the applied research?

Can you add the detailed flowchart of the whole processing to determine CNF including non-detect identification, cloud screening using the VIIRS product, and CACDA?

Can you show the spatial and temporal variations of the detection limit of the CrIS retrievals? In line 86, it is mentioned that the detection limit is ~ 1 ppb. However, the non-detect values are <0.5 ppb in Table 2. Is it appropriate? Don’t you need to take the differences in detection limit into account? Especially, I am concerned about the diurnal differences in the detection limits. I expect that is significant due to the differences in thermal contrast between daytime and nighttime.

The evaluations of the non-detect values are performed only in North America in Sect. 4. Are they appropriate in the other areas? Aren’t there any surface observations other than North America?

Specific comments:

Line 96: Please explain in detail how are the non-detects identified in the CFPR processing.

Table 1: How did you determine smoke plumes when CNF=2? Did you use the VIIRS data?

Fig 9 and 10: Please add the seasons for each panel.

Line 267: Please add space between “>=” and “4”.

Line 396: The cloud screening with COD > 1 should be stated here. I understood that the pixels with COD > 1 were used in Figure A2 but not in Figure A3. Is it right? Please add the detailed treatment of cloudy pixels.

Figure A2: Is this based on the VIIRS product? Details of this analysis should be added. They should differ spatially and temporally because the BT differences are affected by not only optical thickness but cloud height.

Line 416: Please describe the details of the analysis. Did you take into account the differences in temperature profile or water vapor continuum?

Reviewer 2 Report

The manuscript describes a method of estimating missing values of NH3 concentrations derived from CrIS SNPP products. In general, the manuscript is worthy of publication as the topic is of significant interest and is of international interest.  The writing is however in need of simplification as there are numerable sentences that have 3, 4 or 5 clauses- making them very hard to read.  Other concerns include:

35-37: Other approaches to below MDL values is the use of ½ MDL. Perhaps include this as another approach and explain why it is not used

48-49, 84-86: What are the other sources of signal at the 967.5 cm-1 band that could contribute to error of estimation? 

81: how does the measurement time ‘enhance sensitivity in the boundary layer’. Explain

91-93: what is described as ‘background’ may be source areas in which distinct sources are not emitting significantly around that time or being greatly dispersed by turbulence: 14 km pixels are far bigger than most distinct sources.

105-113: Use labels in Fig 1 since many readers will not know the indicated regions which are hard to discern with the cloud cover.  This paragraph should be in Section 2.2

125: CrIS resolution should be stated in section 2.1

136-158: I do not think that most NH3 monitoring is being done in source regions. Although this certainly depends on the magnitudes and scales considered a source-  Does one point source in 14 km make it a source region? Define a non-source region.

212-215: Use labels in Fig 2 since many readers will not know the indicated regions

282-292: Figure 6 is labeled a,b,c,d; use these instead of ‘lower left’ etc.

293-329: It is unclear where Non-detect estimates across the entire globe are coming from. Since the sources identifies across the globe vary widely in relative importance, it is questionable if NA non-detect values (which largely represent advected NH3 from discrete sources) can be applied in other portions of the world where the dominant NH3 sources differ from the NA dominant sources in magnitude, influence of T, and seasonality.   This should be discussed.

Fig. 8: This should be presented in section 3.  

Reviewer 3 Report

The authors present a methodology for quantifying non-detectable values of NH3 in retrievals from CrIS to improve the quality of gridded Level 3 products. They coined the term “non-detects” to describe the portion of background NH3 values that fall below the detection level of CrIS. It is a novel approach with some potential in satellite retrieval systems. This paper, however, does not present well as a scientific document. I have had to reread many sections, even go back to the beginning after learning new information much later in the paper. I encourage the authors to rework and resubmit and hope my review below provides the necessary guidance.

In-line review

The introduction needs serious work. The authors fail to communicate the scientific context of this work and have very few references to other observations systems that deal with the same issue in different ways. They do make broad statements about existing procedures but never substantiate these with appropriate citations. The goal and scope of this work is also not clear from the introduction.

Lines 33 to 35: I have suggestions for the first few sentences of this paper to add clarity.

“Measurements from any instrument have a minimum observable limit. Any values that are below this limit cannot be detected, which we refer to as “non-detects”. The existence of these non-detects complicates statistical analysis and should be excluded from averaged datasets”

It will help the reader if the authors specify that “non-detects” is a new term they coined to describe unmeasurable quantities. I have never before come across this term, but the theory and practice of handling non-detectable values (i.e., the portion of the observable environment that falls in the null space of the measurement) is vast and interesting. The way this paper reads currently creates the impression that the authors are unaware of this rich body of existing work.

Lines 42-43: Sentence should read “Non-detects in trace gas measurements…” not “non-detects in trace gasses”.

Lines 47-48: “A pixel-level non-signal CrIS measurement…” This needs to be reworked. There is no such thing as a non-signal CrIS measurement.

Line 49: I do not know of any CrIS clear-sky “atmospheric signal” that is “mainly a function of ammonia concentrations”. Ammonia is a minor gas species. CrIS measurements, in general, will always measure at the very least atmospheric temperature (as a function of CO2 absorption) and water vapor under any type of clear-sky condition. I suggest the authors specify which part of the CrIS spectrum they refer to here, and to add a few sentences about the range of observables in this spectral region. Can CrIS channels ever be “mainly a function of ammonia concentrations”? I.e., is the signal ever that large?

Line 56: Sentence should read: “…assigned representative, or survival, values”. Also, I’m not familiar with the term “survival values”. I can derive its meaning, but I wonder if this term adds any real clarity here. This is another new term that I have not come across in the scientific literature of satellite inversion methods. It will help the reader if the authors spend a sentence to clarify and justify the use of this term.

Line 69: “NH3 v2 band, around 967.5 cm-1” The CrIS instrument has hundreds of narrow spectral channels, not just a few broad spectral bands. I would like to see the authors specify the spectral range and include an estimate of the number of channels sensitive to NH3. This will give the reader a better idea of the scope of the signal-to-noise the authors describe here and clarify how the CFPR can achieve ~1 DOF for NH3.

Line 72: Add a reference for the CrIS instrument.

Line 80: “NH3 spectral region”, which is what exactly?

Line 80: “~1600” units?

Line 81: “high thermal contrast associated with the early afternoon overpasses” This not true in general for all CrIS measurements across the globe. I suggest the authors clarify or limit their focus to a specific region.

Line 107: Figure 1 does not help the reader distinguish between “clouds” and “smoke”. Rework.

Line 110: “below the detection limit.”, which is what?

Line 138: “Ideally, continuous, in-situ surface ammonia observations … are used.” I assume the authors mean continuous in time, not space? And why are continuous observations ideal? Why not just an observation within a reasonable timeframe of a CrIS overpass?

Line 140: “continuous/instantaneous” Which one do the authors mean? An instantaneous measurement is a single moment in space-time, and thus the opposite of a continuous series of measurements.

Line 146: The authors give geographic coordinates for the CAPMoN site (to the 6th decimal), but omit coordinates for the two other sites. I suggest adopting a consistent approach to referencing these sites. Also, is it relevant that one is “suburban” and the others “rural”?

The 1 x degrees of freedom (DOF) the retrieval algorithm achieves for NH3 retrievals from CrIS appears high to me. As the authors know, DOF is a measure of the signal-to-noise (SNR) of the retrieval system and depends on how various inversion parameters are defined. The degrees of freedom of a retrieved variable is not an absolute quantity inherent in the CrIS measurements, but instead presents as a range across different retrieval systems given CrIS information content and retrieval system design. I’m wondering if this effort to stratify non-detectible background NH3 values according to scene temperature would be unnecessary in retrieval systems where the SNR is better constrained. Can the authors comment on this? The authors state that this method they present here can be applicable to other satellite products, but I am not convinced that this is true.

Line 149: “determine non-detects values…” Awkward sentence construction. Rephrase.

Line 150: “To evaluate accounting for the non-detects…” Awkward sentence construction. Rephrase.

Line 170: Acronym already defined.

Line 180: should be “analyses” not “analysis”

The caption for Table 1 can be more descriptive and remind the reader which product/procedure these flags refer to.

The term, “survival data” is new to me and I wonder if this is really the best description. Can the authors explain why they chose this term and how it serves this paper better than a more intuitive term such as “background values”.

Line 184: “daily August 12, 2017 FOV”… just one FOV? Also, this is an awkward sentence. Rephrase.

Figures 2 and 3: In describing these figures the authors refer to specific regions on the map, but it is not clear at all. I can see a thin yellow line that I assume to be the border between Canada and the USA, but unless the reader knows the “Hudson Bay” area a-priori or consult a separate map, this is not clear at all. I also question the value of using a VIIRS true color image as background. It makes for a complex map that obscures the primary data source, namely the cloud flag data. It is not easy to figure out what is going on.

Line 199: This is the first time the authors refer to survival data with parentheses. See previous comment. I suggest clearly defining the choice and meaning of this word early on in the paper and then standardizing its use throughout.

Line 205: “The median value … at the CrIS overpass time…” Previously the authors stressed the importance of continuous in-situ measurements. I am unsure how one can calculate the median value of a point-source measurement at a specific time. Can the authors clarify this?

Line 207: “Then the average value of the median surface NH3 concentration from all three stations…” How much do these median values vary across the three stations? In general, an average of just three values is not statistically robust.

Table 2: This table presents the NH3 values (in parts per billion) the authors employ in this method as a function of 5ËšC temperature bins. All NH3 values are below 1 ppb (a very small amount, given what is measurable with top of atmosphere infrared measurements) and to the fourth decimal. I struggle to accept that this classification has any real value, because the difference between one temperature bin and the next one does not result in a meaningful difference in NH3 mixing ratios. Take this example for temperature bins [-10 to -5] and [-5 to 0], where the difference in NH3 mixing ration is 0.015 ppb. Does such a small amount really make meaningful difference? And if so, how can the authors know that a difference of 0.015 ppb is real, given how difficult it is to know NH3 globally?

Does the NH3 retrieval algorithm the authors employ here retrieve NH3 in mixing ratio units? Can the authors give a basic introduction of the NH3 retrieval algorithm they used here to help the reader understand how it functions? I see there are no values reported over oceans, for example. Is this an artefact of the retrieval algorithm?

Line 277: “Non-detect conditions occur on a per observation basis” Is this not just a different way of saying that CrIS observational skills vary with temperature, which is an inherent capability of infrared measurements.

Line 341: “post-retrieval processing approach” This is the first time the authors state that this is a post-processing step. It is confusing because throughout the text I thought this was part of the retrieval algorithm. The fact that this is a post-processing step should be communicated clearly at the very beginning to help the reader better understand.

This concluding paragraph added new information that would have served the reader much better had it been in the introduction. It does also compound my confusion. Do the authors mean that the NH3 retrieval algorithm does not retrieve values where SNR<1, and it is only for these cases that this post-processing step should be applied? I.e., should this correction not be applied to retrievals that actually exist? And why?

Broader questions

-       It will help a great deal if the authors add a paragraph to the introduction to give the reader some background about global NH3 distributions. This will also help clarify why variation in very small background values (e.g., 0.015 ppb) matters.

-       At times I couldn’t help but ask: So what? What does it matter if a FOV has 0.5ppb versus 1.0ppb? In my experience, one can change any number of things in the retrieval algorithm that will manifest as a change in aggregated datasets, especially when that aggregation is across 9 years. Are the differences between including/excluding non-detects similar for daily gridded data? I doubt it.

-       Would this post-processing step be necessary if the retrieval algorithm simply had a better NH3 a-priori?

-       Should this post-analysis bias correction of the retrieval product be applied to the Level 2 products before distribution to the general public, or should this method only be applied when generating Level 3 products?

-       Does this post-processor basically add values where the retrieval algorithm could not retrieve NH3? If so, why not include oceans then?

-       How do the authors know that these non-detects are due to the measurements not detecting the quantities, versus the retrieval algorithm introducing artifacts?

-       If very low concentrations of NH3 is not detectable from CrIS measurements, and the retrieval algorithm only retrieves NH3 where SNR >1, why is it necessary to substitute data gaps with generalized values from 3 isolated in-situ sites? Why is it not sufficient to simply know where the source regions are? I guess it will help justify this work if the authors communicate their target user base. s

-       Does the inclusion of non-detects help improve the accuracy of NH3 source regions?

-       How is this method applicable to other products as the authors state in this paper?

Round 2

Reviewer 1 Report

Regarding the first major comment:

Where is the corresponding part in the manuscript? Please add the statements in the manuscript.

Regarding the second major comment:

I understood that the changes in detection limits were considered for each pixel because they are determined by using the ammonia signal and the spectral noises. But they are not taken into account to estimate the non-detect values. My question is why the surface temperature is the only explanatory variable of the non-detect value. Even if the surface temperature is the same in the pixels under the detection limits, the NH3 concentrations can differ depending on the detection limit. I understand that there are not enough data to evaluate the relation between the values of Table 2 and the detection limits. Therefore, I recommend adding the discussions or comments about that in Section 3.2 or 6.

Reviewer 3 Report

I would like to commend the authors for their thorough response to all reviewer concerns. This paper now reads much better. I do, however, still have a few concerns as detailed below.    The line numbers are missing in this rework of the paper, so I hope these comments make sense.    In general and throughout the text, I encourage the authors to be more specific about their references to CrIS measurements/products. It is not always clear whether they mean the Level 1B radiance, the retrieval or the post-processed product.    Page 1: "When averaged over regions or periods, not accounting for these non-detects leads to high biases"  Is this generally true? This is a strong and confusing opening statement. As reader, I am left wondering how this can happen when the measurements themselves have no skill in detecting NH3 in the first place. I'm not convinced this is the case for all retrieval algorithms and all trace gas products. Maybe this statement only applies to certain Level 2 products based on assumptions made in the retrieval algorithms, e.g., such as using static error covariance matrices that does not adjust according to measurement skill (information content) or where high-biased static a-priori estimates/errors are used? Or do the authors mean this applies generally to all trace gas satellite products, even CO and O3? I am not so sure.    Page 1: "A number of procedures for handling non-detects" The list of examples the authors give does not acknowledge the operational CrIS products at both NOAA and NASA. This omission that does not make sense, given that this paper focuses on CrIS retrievals and the Level 2 product they use depend on the NOAA CrIS product. Both NUCAPS (STAR NUCAPS team 2021) and CLIMCAPS (Smith & Barnet 2019, 2020) are all based on the AIRS Science Team methodology and run on CrIS measurements from SNPP and JPSS-x via NOAA CLASS and NASA GES DISC. These use singular-value-decomposition at run-time to quantify measurement information content for target species so that it can address this very problem, namely that SNR vary from scene-to-scene and sometimes is very low for a target species. These operational systems dynamically adjust optimal estimation weighting (and thus the SNR) to avoid retrieving quantities from measurements when the measurements have no sensitivity to the target species at a give scene. While these do not include NH3 in their list of retrieved gaseous species, these systems do deserve to be referenced as an alternative approach to dealing with the issue at hand in this paper.    And then, as I'm thinking through this work described here, I'm wondering if the authors would observe a high bias for products where non-detects are assigned zero values. It brings me right back to my first comment. Is a high bias expected for all products?    Page 2: "when the absolute value of the signal-to-noise ratio (SNR) of the measurement falls below 1 It will help the reader a great deal if the authors define what they mean by SNR, especially since they're so specific about a threshold of 1. The authors did explain this in their response to a reviewer question, but I strongly encourage the authors to add this explanation to the main text.    Page 2: "Given the importance of NHit is important to..."     Page 2: "Inaccuracies in NH3 measurements" I would replace the word "measurement" with "retrieval" here because, by definition, the CrIS instrument does not measure NH3 when NH3 falls below the detection limit. It is the retrieval systems that introduce these high-biased NH3 values, not the CrIS instrument measurements.    Page 2: "...can cause problems...". Remove excessive "and"    Page 2: "Satellite data are unable to detect..." This statement is somewhat vague and misleading. Given their focus on CrIS, I suggest the authors stick to highlighting the shortcomings of infrared (IR) satellite measurements. And then, I would argue that spatial resolution is not the primary factor that limits IR observability of NH3 in the boundary layer. It is spectral resolution coupled with the fact that IR sounding capability depends on temperature, and specifically temperature contrast and lapse rate. Moreover, the sources of NH3 the authors list tend to exceed 15 km^2 in range, so a smaller measurement footprint would not hold specific benefit here. Perhaps a smaller footprint would help find more holes through partly cloudy atmospheres and thus offer more cloud-free retrievals, but that is a secondary benefit as far as I understand this.    Page 2: "low value NH3 value" remove duplicate word   Standardize font on Page 3   Page 3, Section 2.1, first paragraph: As the authors know, CrIS on SNPP was originally downlinked at nominal spectral resolution (NSR) and late in 2015 restored to its full spectral resolution (FSR). It will satisfy a reader familiar with this to add a sentence here stating that the long-wave CrIS band was not affected by this.   Page 3: "The high CrIS SNR (~1600)...". I think the authors use the same definition of SNR as on page 1? It will avoid confusion if the authors can be more specific so that the reader is not left with the impression that CrIS SNR can vary from ~1600 to <1. Perhaps the authors mean that for all spectral channels, the CrIS SNR is ~1600 and for the 10 channels used in the NH3 retrieval, SNR can be <1?    Page 3: NUCAPS: Spell acronym out in full on first use.    Page 3: "The CrIS NH3 detection limit..." This sentence is confusing. Perhaps rephrase as follows "...and on the surface temperature and vertical atmospheric temperature". And what do the authors mean by "ranges from ~0.3 to 1.0 ppbv"? This clashes with the sentence on page 2 "measure values ranging from 0.1 ppbv to 100 ppbv".    Page 4: "where the lowest concentration values of CrIS have..." This is an important sentence and sheds light on the source of the high bias earlier reported. But I am confused. CrIS measures emitted radiance, not trace gas concentrations. Do the authors mean that the CrIS radiances are high biased in the NH3 absorption region compared to ground-based FTIR measurements?    Figures 1–3: These figures continue to give me pause. I'm afraid that in attempting to fix issues identified during the first reviewing cycle, the authors have amplified them. I accept that the authors wish to keep these figures and use them to help highlight certain issues. Two suggestions: (i) draw a box around the area the authors wish to highlight with a thick off-white line, and/or (ii) zoom in to the area of interest.    Page 6: "...some FOVs with above detection limit NH3 values..." Awkward sentence. Maybe rephrase as follows: "some FOVs with NH3 values above detection levels"   Table 1: Specify which "CrIS dataset" the authors refer to here, namely the CFPR Level 2 retrieval product. Some readers may interpret this as a Level 1B categorization.    Section 5: "Level2" should be "Level 2". Same for "Level3". There are a number of such spacing issues throughout the paper that the authors can correct in the final proofread.    I would like to encourage the authors to address these nits with a few basic changes in the sentences. Overall, this is an interesting method with potential value to other products.    References STAR NUCAPS Team (2021). NOAA Unique Combined Atmospheric Processing System (NUCAPS) Algorithm Theoretical Basis Document. National Oceanic and Atmospheric Administration (NOAA) National Environmental Satellite Data and Information Service (NESDIS) V3.1. https://www.star.nesdis.noaa.gov/jpss/documents/ATBD/ATBD_NUCAPS_v3.1.pdf

Smith, N., & Barnet, C. D. (2019). Uncertainty Characterization and Propagation in the Community Long-Term Infrared Microwave Combined Atmospheric Product System (CLIMCAPS). Remote Sensing11(10), Article 10. https://doi.org/10.3390/rs11101227
